# Defective internal allosteric network imparts dysfunctional ATP/substrate-binding cooperativity in oncogenic chimera of protein kinase A

Cristina Olivieri [1,8], Caitlin Walker [1,8], Adak Karamafrooz[1,8], Yingjie Wang[1,2,7], V. S. Manu[1], Fernando Porcelli [3], Donald K. Blumenthal [4], David D. Thomas [1], David A. Bernlohr [1], Sanford M. Simon[5], Susan S. Taylor [6] & Gianluigi Veglia [1,2✉]

An aberrant fusion of the *DNAJB1* and *PRKACA* genes generates a chimeric protein kinase (PKA-C[DNAJB1]) in which the J-domain of the heat shock protein 40 is fused to the catalytic α subunit of cAMP-dependent protein kinase A (PKA-C). Deceivingly, this chimeric construct appears to be fully functional, as it phosphorylates canonical substrates, forms holoenzymes, responds to cAMP activation, and recognizes the endogenous inhibitor PKI. Nonetheless, PKA-C[DNAJB1] has been recognized as the primary driver of fibrolamellar hepatocellular carcinoma and is implicated in other neoplasms for which the molecular mechanisms remain elusive. Here we determined the chimera's allosteric response to nucleotide and pseudo-substrate binding. We found that the fusion of the dynamic J-domain to PKA-C disrupts the internal allosteric network, causing dramatic attenuation of the nucleotide/PKI binding cooperativity. Our findings suggest that the reduced allosteric cooperativity exhibited by PKA-C[DNAJB1] alters specific recognitions and interactions between substrates and regulatory partners contributing to dysregulation.

[1] Department of Biochemistry, Molecular Biology, and Biophysics, University of Minnesota, Minneapolis, MN, USA. [2] Chemistry, University of Minnesota, Minneapolis, MN, USA. [3] DIBAF - University of Tuscia – Largo dell' Università, Viterbo, Italy. [4] Department of Pharmacology and Toxicology, University of Utah, Salt Lake City, UT, USA. [5] Laboratory of Cellular Biophysics, Rockefeller University, New York, NY, USA. [6] Department of Chemistry and Biochemistry and Pharmacology, University of California at San Diego, La Jolla, CA, USA. [7] Present address: Shenzhen Bay Laboratory, Shenzhen, China. [8] These authors contributed equally: Cristina Olivieri, Caitlin Walker, Adak Karamafrooz. ✉email: vegli001@umn.edu

Fibrolamellar hepatocellular carcinoma (FL-HCC) is a rare and aggressive form of liver cancer that predominantly affects young patients without underlying cirrhosis or disease[1]. Treatment is limited, as FL-HCC does not respond to chemotherapy[2,3], and surgical excision remains the only therapy[4,5]. Sequencing the tumor genomes of FL-HCC patients identified a recurrent chimeric gene[6–9] that originates from a ~400 kb deletion in chromosome 19. This deletion causes an in-frame fusion of exon 1 from the *DNAJB1* gene, which encodes a member of the heat-shock protein 40 family, with exons 2–10 of the *PRKACA* gene, encoding for the catalytic subunit of cAMP-dependent protein kinase A (PKA-C)[7]. The resulting chimeric enzyme (PKA-C$^{DNAJB1}$) is fully functional and comprises 405 residues with 69 amino acids of DNAJB1 (J-domain), replacing the first 14 residues of the N-terminal αA-helix of PKA-C[7]. The N-terminal deletion of part of the αA-helix prevents essential post-translational modifications such as N-myristoylation, deamidation, as well as phosphorylation at Ser10, which are responsible for the spatiotemporal localization of the kinase[10,11].

The fusion of the J-domain does not alter the structure of the catalytic core of PKA-C$^{DNAJB1}$, which remains virtually identical to the wild-type kinase (PKA-C$^{WT}$). Instead, the αA-helix of PKA-C$^{DNAJB1}$ is extended with the J-domain tucked under the large lobe of the enzyme (Fig. 1a)[12]. Moreover, PKA-C$^{DNAJB1}$ retains the ability to form stable holoenzymes with all regulatory (R) subunit isoforms, although with non-canonical arrangements, and more importantly, responds to cAMP activation[13,14]. In vivo enzymatic assays, as well as conventional in vitro coupled assays, have shown that the kinetic parameters of PKA-C$^{WT}$ and PKA-C$^{DNAJB1}$ are similar[7,13,15]. Furthermore, PKA-C$^{DNAJB1}$ binds the endogenous inhibitor PKI (both full-length and derived peptides) with an affinity comparable to wild-type[15].

Despite little differences conferred to the structure and activity of the kinase by the fusion of the J-domain, in vivo studies have shed light on the signaling changes induced by the chimeric fusion.

Recently, Scott and co-workers suggested that the J-domain is able to recruit Hsp70, stabilizing the chimeric fusion protein[16]. Together with the higher expression levels due to the *DNAJB1* gene[7], the enhanced stability of the enzyme in complex with Hsp70 may explain the dominant expression in FL-HCC cells. Furthermore, these researchers showed that the phosphoproteomics of PKA-C$^{DNAJB1}$ is different from that of the wild-type, implying that the J-domain skews the phosphorylation profile of cells toward alternative pathways[16]. According to this study, the J-domain is directly implicated in the oncogenicity of PKA-C$^{DNAJB1}$. However, other evidence shows that the kinase domain is also required for tumorigenesis[17]. To date, it remains unclear how this aberrant kinase contributes to the progression of the disease.

A prominent feature of PKA-C is the alternation of positive and negative binding cooperativity that drives the enzymatic cycle[18]. Positive binding cooperativity induced by ATP-binding enhances the affinity for the substrate, whereas negative binding cooperativity of ADP facilitates the release of the phosphorylated product. Since binding cooperativity is essential to signal trans-duction amplification[19], we surmised that the nucleotide/sub-strate-binding cooperativity of the aberrant kinase is disrupted, with implications for phosphorylation profiles and regulation by R-subunits. With this in mind, we investigated local and global responses of the kinase to nucleotide and pseudo-substrate binding using a combination of NMR spectroscopy, isothermal titration calorimetry (ITC), small-angle X-ray scattering (SAXS), and molecular dynamics (MD) simulations. We found that the J-domain fused to the N-terminus of PKA-C dramatically attenu-ates the canonical positive nucleotide/pseudo-substrate-binding cooperativity through alterations in the intramolecular network of communication. Analogous to other pathologic mutants of PKA-C[20], the concomitant loss in binding cooperativity and intramolecular communication may render highly cooperative events, such as regulation, dysfunctional, contributing to disease progression.

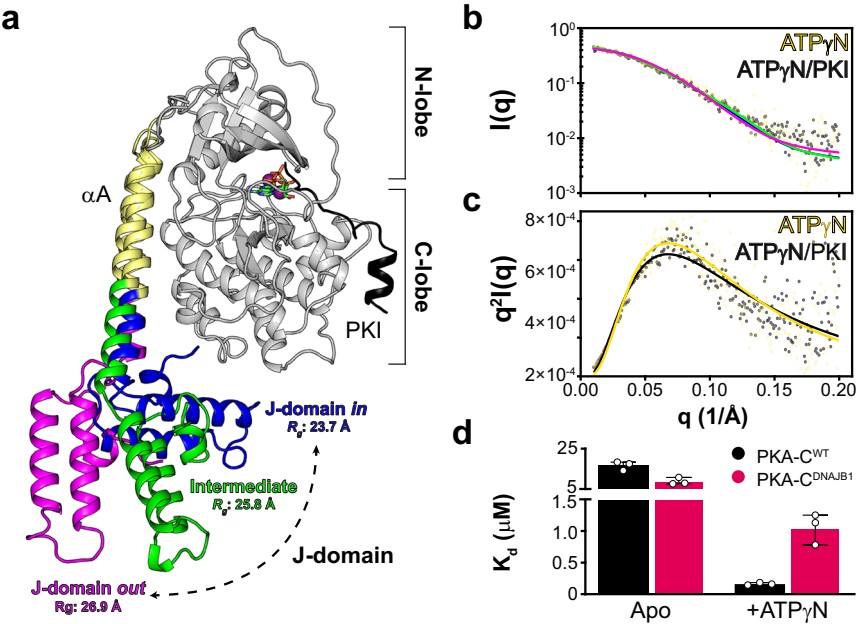

**Fig. 1 Multiple conformations adopted by the dynamic J-domain alters allosteric binding cooperativity of PKA-C. a** Overlay of selected snapshots of PKA-C$^{DNAJB1}$ from the MD trajectories highlighting the ensemble of conformers sampled by the J-domain. **b** SAXS profiles of PKA-C$^{DNAJB1}$. Continuous lines show the structural fitting of the experimental SAXS data with the closed (J-domain *in*, blue), intermediate (green), and extended (J-domain *out*, magenta) conformations for binary (ATPγN-bound) and ternary (ATPγN/PKI-bound) forms. The conformations are extracted from the MD trajectories. **c** Corresponding Kratky plot of PKA-C$^{DNAJB1}$ bound to ATPγN and ATPγN/PKI. **d** Bar graph showing the dissociation constants ($K_d$) calculated from the ITC titration curves of PKI$_{5–24}$ binding to the apo and ATPγN-bound forms of PKA-C$^{WT}$ and PKA-C$^{DNAJB1}$ (see also Supplementary Table 2).

## Results

### The dynamic J-domain of PKA-C$^{DNAJB1}$ interconverts between *in* and *out* conformations.

Previous X-ray crystallographic studies[12] have shown that the core of PKA-C$^{DNAJB1}$ adopts a conformation similar to the wild type. The catalytic domain in complex with an ATP analog and a pseudo-substrate peptide is virtually superimposable to the structure of the wild-type enzyme[12,21], whereas the J-domain of the chimera is tucked tightly under the large lobe of the enzyme (Fig. 1a). In contrast, our computational and solution NMR studies suggest that PKA-C$^{DNAJB1}$ spans a large ensemble of conformations, with the J-domain undergoing fast re-orientational dynamics relative to the core of the enzyme[22]. To provide orthogonal validation, we compared the SAXS profiles of PKA-C$^{WT}$ and PKA-C$^{DNAJB1}$. For both binary (ATPγN-bound) and ternary (ATPγN/PKI-bound) complexes of PKA-C$^{DNAJB1}$, we fit the SAXS profiles with the compact (J-domain *in*), intermediate, and extended (J-domain *out*) conformations extracted from the trajectories of MD simulations (Fig. 1b, c). We found all structures fit well with the SAXS profile with $\chi < 1.0$ in the low $q$ region ($q < 0.2$ Å$^{-1}$). However, the radius of gyration ($R_g$) increases from 23.7 to 26.9 Å, indicating that the J-domain can dislodge from the kinase core and adopt an extended conformation. In contrast, the flexible N-terminus of PKA-C$^{WT}$ is much smaller and does not influence the $R_g$ of the enzyme (21.7 Å) for both binary and ternary complexes (Supplementary Fig. 1). The addition of PKI to nucleotide-saturated PKA-C$^{WT}$ and PKA-C$^{DNAJB1}$ does not change the SAXS profiles, suggesting that PKI binding does not alter the overall size and shape of the two kinases.

### The J-domain of PKA-C$^{DNAJB1}$ reduces the nucleotide/substrate degree of binding cooperativity ($\sigma$).

Previous studies from our group have shown that PKA-C$^{WT}$ behaves like a K-type enzyme, exhibiting positive binding cooperativity between nucleotide (ATPγN) and pseudo-substrate (PKI$_{5-24}$). The latter property can be reduced by mutations, while maintaining a virtually constant maximal rate[20,23,24]. Thus, we sought to understand how this canonical positive binding cooperativity is altered in PKA-C$^{DNAJB1}$, using isothermal titration calorimetry (ITC). For the binding of nucleotide alone, PKA-C$^{DNAJB1}$ exhibits a fourfold higher binding affinity for ATPγN compared with PKA-C$^{WT}$ ($K_d = 19 \pm 4$ μM versus $83 \pm 8$ μM; Supplementary Table 1 and Supplementary Fig. 2). To determine the degree of binding cooperativity ($\sigma$) for PKA-C$^{DNAJB1}$, we monitored the binding affinity of PKI$_{5-24}$ in the absence and presence of nucleotide. In the absence of nucleotide, PKA-C$^{DNAJB1}$ binds PKI$_{5-24}$ with a twofold higher affinity compared to wild-type ($K_d = 9 \pm 2$ μM versus $17 \pm 2$ μM; Fig. 1d and Supplementary Fig. 2). In contrast, upon saturation with nucleotide, the PKA-C$^{DNAJB1}$/ATPγN complex binds PKI$_{5-24}$ with a sevenfold decrease in binding

affinity ($K_d = 1.1 \pm 0.2$ μM versus $0.16 \pm 0.02$ μM). Together, these data show that the addition of the J-domain causes a sizeable decrease (13-fold) in nucleotide/pseudo-substrate-binding cooperativity (Supplementary Table 2).

To evaluate the effect of the dramatic reduction in binding cooperativity on the ability of PKA-C$^{DNAJB1}$ to perform phosphoryl transfer, we used steady-state coupled enzyme assays. Specifically, we evaluated the catalytic efficiencies of PKA-C$^{WT}$ and PKA-C$^{DNAJB1}$ toward the standard peptide substrate, kemptide, physiological substrate, CREB, and a known hyperphosphorylated substrate identified by Scott and co-workers, KSR1[16] (Fig. 2a). Though the substrates displayed different maximal velocities ($V_{max} = 0.21-0.35$ μM/s) and Michaelis constants ($K_M = 29-56$ μM), PKA-C$^{DNAJB1}$ consistently displayed higher $V_{max}$ values resulting in slightly higher catalytic efficiencies ($k_{cat}/K_M$) compared with wild-type for all substrates tested (Fig. 2b, Supplementary Table 3, and Supplementary Fig. 3). Although the binding cooperativity is affected by the presence of the J-domain, these assays with peptides encompassing the recognition sequences of different substrates demonstrate that the kinetic parameters are only marginally altered as reported previously[15].

### The J-domain disrupts the internal allosteric network of PKA-C$^{DNAJB1}$.

To determine the origin of the decreased allosteric binding cooperativity, we analyzed the structural response of the kinases to ligand binding using solution NMR spectroscopy. Specifically, we mapped the amide fingerprints of both enzymes using [$^1$H, $^{15}$N]-TROSY-HSQC experiments[25] upon addition of nucleotide and PKI$_{5-24}$ (Supplementary Fig. 4). The amide fingerprints of PKA-C$^{WT}$ and the PKA-C$^{DNAJB1}$ kinase core are almost superimposable, reflecting their identical structures. This similarity enabled us to transfer the previously assigned PKA-C$^{WT}$ amide resonances[23] to the PKA-C$^{DNAJB1}$ spectrum (Supplementary Fig. 4a). To assign the remaining resonances of the J-domain, we expressed the 69-residue construct (DNAJB1$_{1-69}$) uniformly labeled with $^{15}$N and $^{13}$C in *Escherichia coli* bacteria and assigned the resonances in the [$^1$H, $^{15}$N]-TROSY-HSQC spectrum using the classical *out-and-back* triple resonance experiments (Supplementary Fig. 5). A total of 340 amide resonances were assigned over the 388 expected. As previously reported[26], the DNAJB1$_{1-69}$ domain folds autonomously, and its resonances overlay with those of the J-domain fused to PKA-C$^{DNAJB1}$ (Supplementary Fig. 4b). To analyze the kinases response to ATPγN and PKI$_{5-24}$ binding, we followed the $^1$H and $^{15}$N chemical shift perturbations (CSPs) of the amide fingerprints for the two kinases. We found that the fingerprints of both PKA-C$^{WT}$ and PKA-C$^{DNAJB1}$ show similar CSPs upon binding ATPγN (Supplementary Fig. 6a). However, residues of PKA-C$^{DNAJB1}$ encompassing the Gly-rich loop, activation loop, αF-helix,

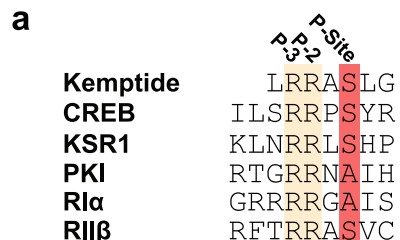

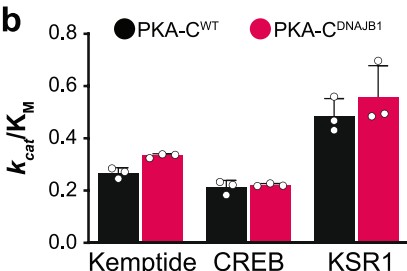

**Fig. 2 Attenuation in binding cooperativity does not influence phosphoryl transfer. a** Comparison of the substrate recognition sequences of substrates Kemptide, CREB, and KSR1, regulatory subunits, RIα and RIIβ, and endogenous inhibitor, PKI. **b** Bar graph displaying the catalytic efficiencies ($k_{cat}/K_M$) of PKA-C$^{WT}$ and PKA-C$^{DNAJB1}$ towards Kemptide, CREB, and KSR1. Errors were calculated from triplicate measurements. All kinetics parameters are summarized in Supplementary Table 3 and shown in Supplementary Fig. 3.

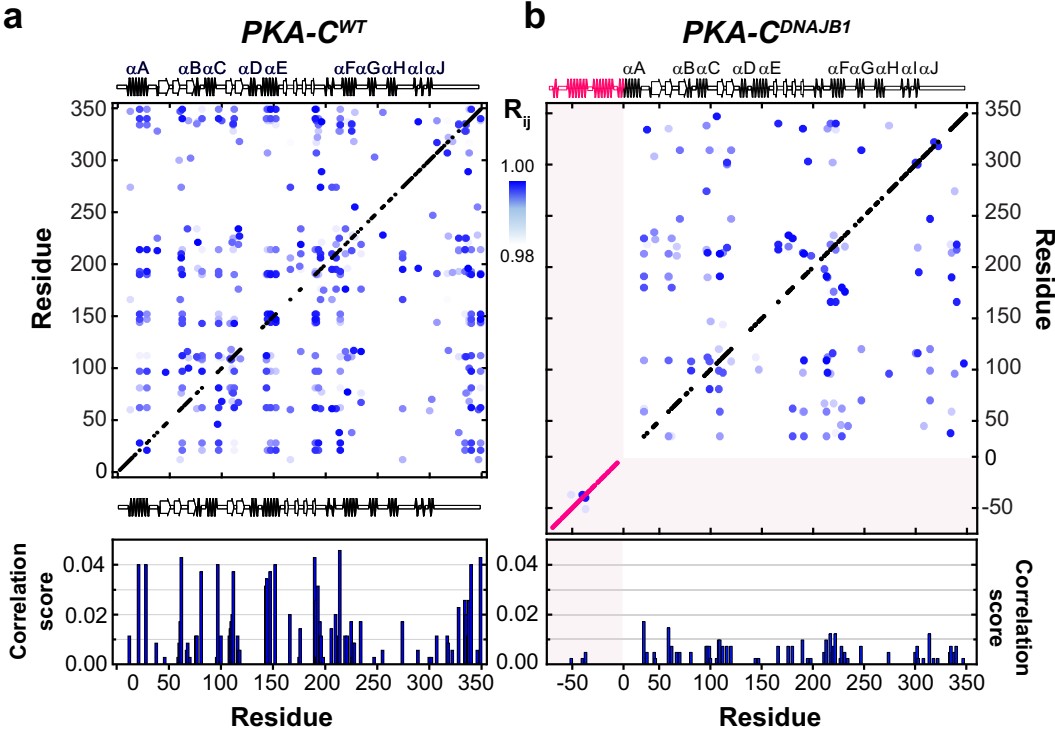

**Fig. 3 Defective internal allosteric network of PKA-C^DNAJB1.** Chemical shift correlation matrices for **a** PKA-C^WT and **b** PKA-C^DNAJB1. The cross-peaks represent pairs of residues with the absolute value of the Pearson's coefficient ≥0.98. The secondary structures for the two kinases are reported above and below the matrix. The bar graphs shown below each matrix report the average correlation scores per residue (see Methods section).

αG-helix, and the C-terminal tail show greater CSPs compared with PKA-C^WT. Notably, these domains undergo the largest structural rearrangements from the open (apo) to closed (ternary) form. The J-domain is not directly involved in the binding events as it experiences CSPs well below one standard deviation. As for PKA-C^WT, the binding of ATPγN causes broadening of amide resonances throughout the core of PKA-C^DNAJB1, suggesting the presence of conformational dynamics in the μs–ms timescale[27]. Upon binding PKI$_{5-24}$ to the ATPγN-saturated kinase, the CSPs of the kinase cores are similar, with effects propagated from the substrate-binding sites to both small and large lobes (Supplementary Fig. 6b). The J-domain remains structurally isolated from the core as we observe only a few chemical shift changes upon binding PKI$_{5-24}$. As found for PKA-C^WT, PKI$_{5-24}$ binding sharpens the amide fingerprint resonances, indicating that the enzyme is trapped in a well-defined conformational minimum[23]. The linewidths of the J-domain, on the other hand, are unperturbed, suggesting that pseudo-substrate binding does not change its conformational dynamics.

To analyze the internal network of communication of both PKA-C^DNAJB1 and PKA-C^WT, we used CHEmical Shift Covariance Analysis (CHESCA)[28,29]. This method identifies allosteric networks among the protein residues by tracking each residue's coordinated changes of chemical shifts in response to ligand binding or mutations. For a binding process occurring in the fast NMR timescale, the resonances associated with allosterically coupled residues exhibit a correlated linear response. When plotted on a correlation matrix, changes in allosteric networks manifest as variations in the number and extent of inter-residue correlations[29,30]. We analyzed the chemical shift responses of four different forms of the kinase: apo, ATPγN-bound, ADP-bound, and ATPγN/PKI$_{5-24}$bound. The CHESCA matrix of PKA-C^WT shows a coordinated response for many residues spanning the entire enzyme, with a higher density of correlations

in the N-lobe near the nucleotide-binding site and the substrate-binding groove at the interface between the N- and C-lobes (Fig. 3a). These correlations constitute central allosteric nodes necessary for binding cooperativity[18,20,31]. In addition, allosteric cross-talk exists between residues in the C-terminal tail and residues in the αA-helix (K21, K28), αE-helix (R144, A147, L152), activation loop (R190, V191, G193), and αF-helix (G214, G225). In contrast, the matrix of PKA-C^DNAJB1 exhibits an overall attenuation in CHESCA correlations throughout the entire kinase, with a rewiring of the network between nucleotide and substrate-binding sites. Specific structural domains such as the αA-helix (K21, K27), αE-helix (A143, R144, A147, L152), αC-helix (Q96, A97), and C-terminal tail (I335, N340, E349) are more noticeably attenuated (Fig. 3b). The functional relevance of this extensive rewiring of intramolecular communication is apparent when we combined CHESCA with the definition of the structural and functional communities (community CHESCA)[32]. Upon dividing the kinase into nine different communities defined by McClendon et al.[32] (Fig. 4a), each with different structural and functional roles, we observe a clear parallel between experimental data and theory[31,32]. The correlation matrix for community CHESCA for PKA-C^WT (Fig. 4b) shows that structurally adjacent communities in the N-lobe and at the N/C lobe interface (ComA, ComB, ComC, and ComE) are strongly coupled with distal communities (ComH) as they exhibit high correlation coefficients (R$_{X,Y}$). A structural representation of the extent of coupling is reported in Fig. 4d, where cross-talk between the nucleotide-binding (ComA) with the positioning of the αC-helix (ComB), the R-spine assembly (ComC) and the activation loop (ComF) is apparent. Although attenuated, there is a substantial coupling between these communities and the more peripheral communities that have structural roles, such as ComH. The cooperative response to nucleotide and PKI$_{5-24}$ binding involves the entire enzyme, causing coordinated CSPs

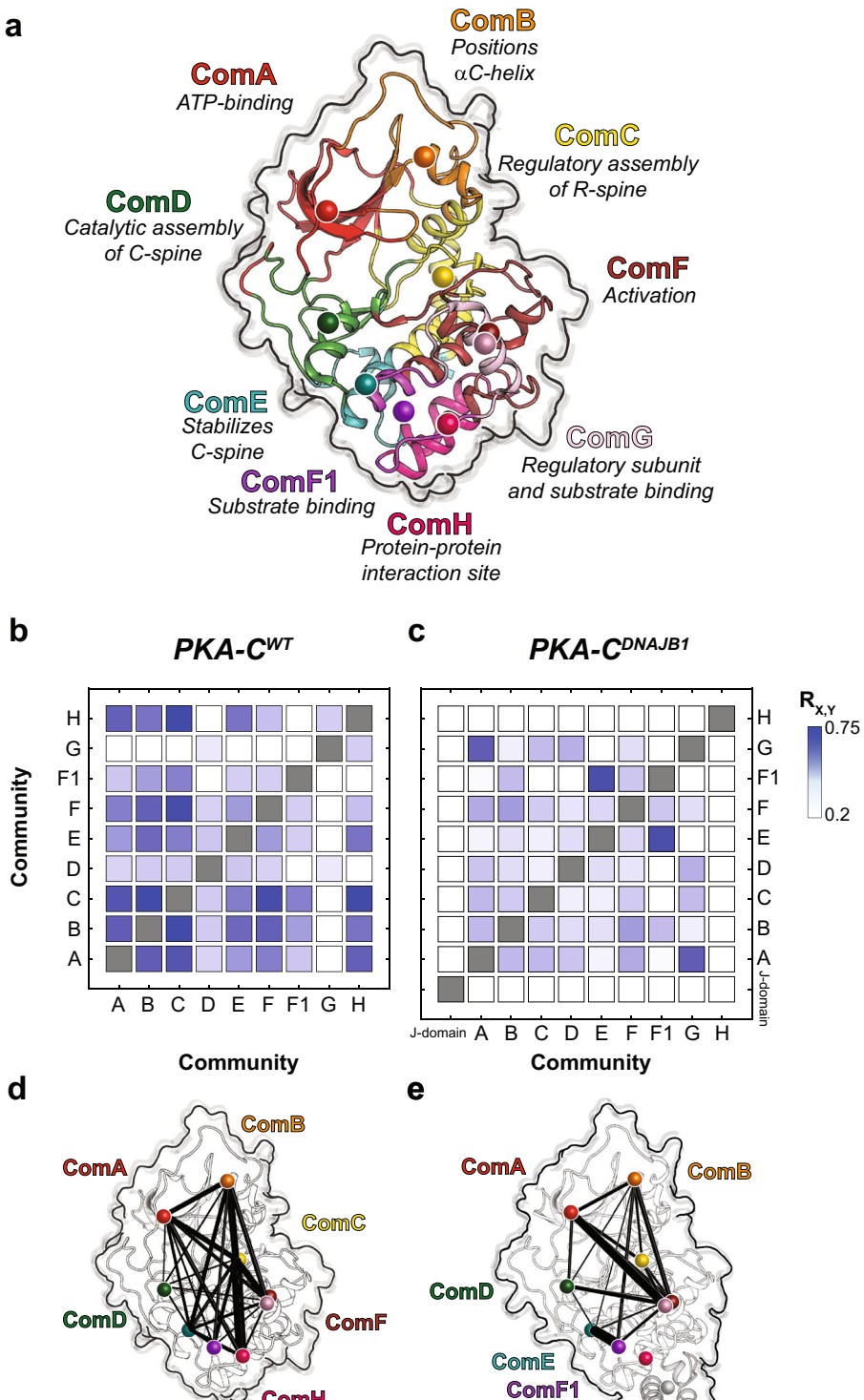

**Fig. 4 Reduced cross-talk among structural communities of chimeric PKA-C<sup>DNAJB1</sup> relative to PKA-C<sup>WT</sup>. a** Structure of PKA-C highlighting all communities and their roles in various functions and regulatory mechanisms. CHESCA community matrices for **b** PKA-C<sup>WT</sup> and **c** PKA-C<sup>DNAJB1</sup> upon binding PKI$_{5-24}$. **d–e** CHESCA community matrices plotted on their corresponding structures. The size of each node is independent of the number of residues it encompasses, meanwhile the weight of each line indicates the strength of coupling between the individual communities.

interspersed throughout its three-dimensional structure. In contrast, the community CHESCA for PKA-C$^{DNAJB1}$ exhibits a dramatic decrease in inter-community coupling (Fig. 4c, e). The coupling network between the ATP-binding site (ComA) and the

αC-helix (ComC) or the activation loop (ComF) is sparse. Notably, the correlations between the N-lobe communities with peripheral communities in the C lobe (ComD and ComE) are either notably reduced or lost. This phenomenon is especially

evident for ComH located near the J-domain, which includes residues that are involved in docking of the R-subunits or kinase binding partners. Other correlations not found in the wild-type are observed between ComG and ComA, ComB, ComD, and ComF; and stronger correlations are observed between ComE and ComF1. Overall, the community CHESCA show that the reduced degree of cooperativity determined thermodynamically corresponds to a decrease in coordinated structural changes upon ligand binding in PKA-C$^{DNAJB1}$.

**Global versus community-specific responses to ligand binding.** Although CHESCA provides an estimate of the correlated response for individual residues (or communities) to ligand binding, the analysis of the amide chemical shifts with CONCISE (COordiNated ChemIcal Shifts bEhavior)[33] provides the probability density (population) of each state along the conformational equilibrium. According to CONCISE analysis, the probability density of the amide resonances for the *apo* form of the enzyme (open state) is clustered around an average principal component (PC) score (<PC>) of ~−1.2 (Fig. 5a)[34]. Upon binding ATPγN, ADP, and PKI$_{5-24}$, the probability densities for the amide resonances of PKA-C$^{WT}$ progressively shift toward more positive <PC> values, with the density for the binary complex (intermediate state) showing a <PC> ~ 0, and the ternary complex an <PC> ~1.3 (closed state). A similar trend is observed for PKA-C$^{DNAJB1}$, though the probability densities are broader, indicating that the amide resonances follow a less coordinated response[33]. Also, the maximum probability density for the intermediate state of PKA-C$^{DNAJB1}$ is slightly shifted toward the closed conformation, whereas the probability density

for the ternary complex is slightly more open than the corresponding wild-type (Fig. 5b). These probability densities or populations of each state can be converted into free energy, and the shift of the maxima from free to bound state along the <PC> axis reflects the free energy of binding[33]. Indeed, the ΔΔG values obtained from ITC data are very similar to those calculated from CONCISE (Fig. 5c, d and Supplementary Table 4). Consequently, the attenuation in the collective response of the amide resonances of the PKA-C$^{DNAJB1}$ is correlated with the reduced extent of binding cooperativity. Interestingly, we found that the two enzymes respond to ligand binding in a community-specific manner (Fig. 6a, b). For PKA-C$^{WT}$, the resonances associated with ComA have the most prominent response upon nucleotide binding. Also, large conformational transitions are observed for ComB, ComC, ComD, and ComE, whereas the remaining communities, located mostly in the C lobe, show only marginal changes, with an <PC1> of ~−0.4. Binding of PKI$_{5-24}$ to the PKA-C/ATPγN complex shifts all the communities toward a fully closed state (<PC1> ~1) (Fig. 6c and Supplementary Fig. 7a). In contrast, the probability densities for ComA, ComB, ComC, ComD, and ComE for PKA-C$^{DNAJB1}$ are shifted to a more closed intermediate state (<PC1> ~0). Moreover, PKI$_{5-24}$ binding to PKA-C$^{DNAJB1}$/ATPγN complex shifts the probability densities of the various communities toward the closed state to different extents (<PC1> ~0.75), with only ComF, ComF1, ComG, and ComH reaching a fully closed state. The remaining communities located in the N-lobe cluster around PC1 ~0.65, adopting a slightly more open conformation (Fig. 6d and Supplementary Fig. 7b). Overall, the J-domain fused to the catalytic core decreases the number of correlated chemical shift responses, i.e., the community of the kinase no longer transition in a correlated

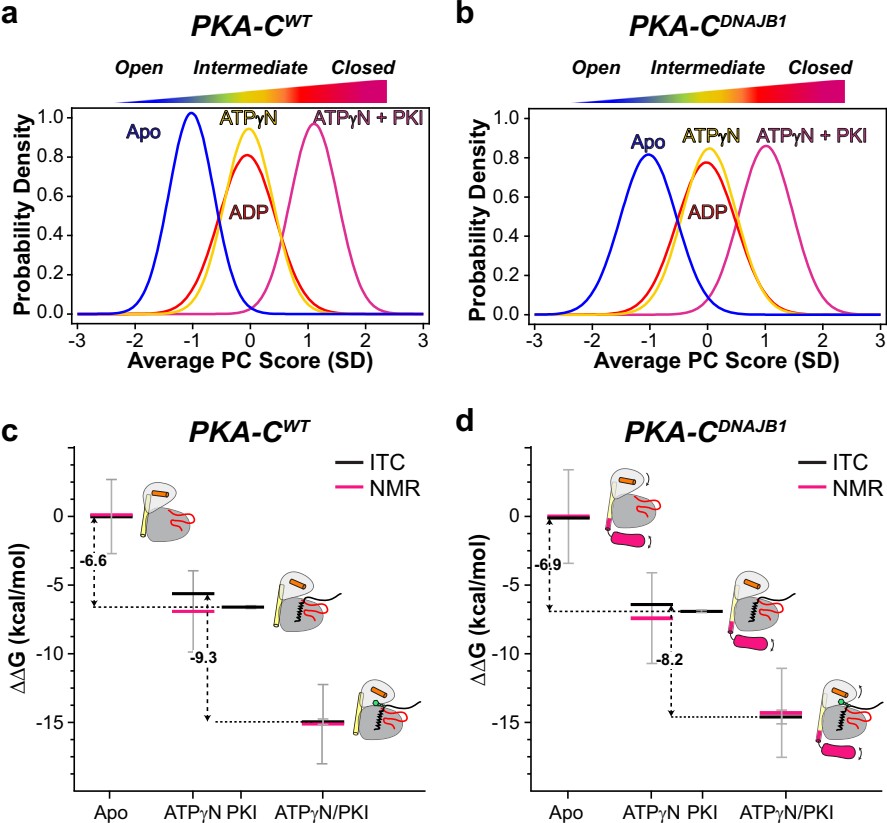

**Fig. 5 Global coordinated response of PKA-C$^{WT}$ and PKA-C$^{DNAJB1}$ to ligand binding.** Coordinated chemical shift changes analyzed with CONCISE for **a** PKA-C$^{WT}$ and **b** PKA-C$^{DNAJB1}$. Changes in the free energy of binding of **c** PKA-C$^{WT}$ and **d** PKA-C$^{DNAJB1}$ to ATPγN and PKI$_{5-24}$. Black lines indicate the values obtained from ITC and pink lines correspond to values obtained from NMR measurements.

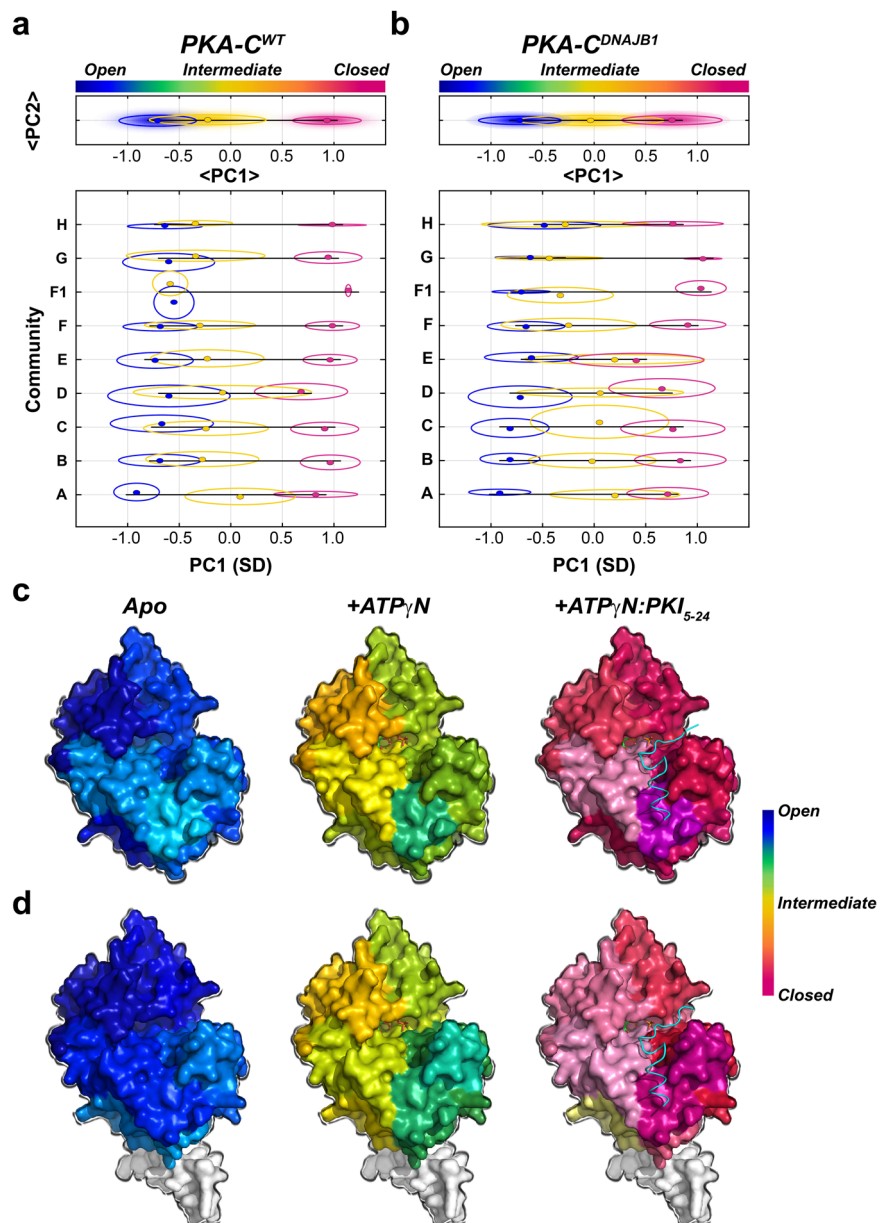

**Fig. 6 Community-specific responses of PKA-C^WT and PKA-C^DNAJB1 upon ATPγN and PKI_{5-24} binding.** CONCISE analysis for each individual community for **a** PKA-C^WT and **b** PKA-C^DNAJB1. The top panel on the graphs shows the average PC1 (<PC1>) overall communities, whereas the bottom panel shows the PC1 scores of each individual community. CONCISE scores of each community mapped onto the surface of **c** PKA-C^WT (PDB: 4WB5) and **d** PKA-C^DNAJB1 (PDB: 4WB7) for each state (apo, +ATPγN, and +ATPγN/PKI_{5-24}).

fashion, and prevents several communities (ComA, ComB, ComC, ComD, and ComE) from reaching a fully closed state, which explains the reduced extent of nucleotide/substrate-binding cooperativity[24].

## Discussion

There is mounting evidence that PKA-C^DNAJB1 is the primary driver of FL-HCC[6–8] and may be involved in other pathologies[35,36]. Despite its link to tumorigenesis, initial structural and functional studies have left more questions than answers regarding the molecular etiology of this rare liver carcinoma. Similar to other oncogenic kinases involved in Cushing's syndrome, the X-ray structure of PKA-C^DNAJB1 does not reveal any structural anomalies in the catalytic core (RMSD < 0.4 Å)[12]. In solution, however, NMR and SAXS have shown that the J-

domain is flexible, assuming both *in* and *out* conformations[22]. The inherent flexibility of the J-domain enables the kinase to form hetero-tetrameric holoenzymes with RIα subunits, although with two alternative conformations[14]. Interestingly, the RIα subunit is overexpressed in carcinoma cells, presumably to "buffer" the PKA-C^DNAJB1 overexpression[6,13,37]. In rare cases of FL-HCC, lack of PKA-C regulation by RIα may result in the progression of the disease[38], like the dysregulation of PKA-C in Carney Complex[39]. These data alone do not adequately explain the aberrant function of PKA-C^DNAJB1.

It is possible, however, that the functional repercussion of the J-domain fusion to PKA-C is multifaceted. First, the deletion of 14 residues in the αA-helix and N-terminus prevents post-translational modifications such as myristoylation, deamidation, and phosphorylation of Ser10. These modifications are essential for intracellular membrane localization[40,41]; therefore, the J-

domain may interfere with the localization of the enzyme in the proximity of natural substrates. Second, the phosphorylation profile of PKA-C[DNAJB1] is considerably different from the wild-type[16]. Third, the presence of the J-domain may interfere with the formation of localization complexes with A-kinase anchoring proteins (AKAPs)[37]. Fourth, PKA-C[DNAJB1] may form stable complexes with Hsp70, with the J-domain acting as a scaffold for this chaperonin[16].

Irrespective of these putative mechanisms, our studies reveal that the fusion of the J-domain to the kinase reduces canonical positive binding cooperativity between nucleotide and substrate[24], and alters the kinases' conformational equilibrium via disruptions of the intramolecular allostery. Indeed, cooperativity has an essential role in macromolecular assembly, regulation, and signal transduction[19,42]. Therefore, the attenuation in allosteric binding cooperativity is likely to be implicated in the altered interactome and phosphoproteome[16]. As the recognition sequences of R-subunits are highly homologous to PKI, the reduced nucleotide/PKI binding cooperativity of PKA-C[DNAJB1] suggests an aberrant regulation of the holoenzyme as well. It is worth noting that a reduction in cooperativity is also a prominent feature of a single mutation in the P + 1 loop of the catalytic subunit (PKA-C[L205R]) in patients affected by Cushing's syndrome[20,43,44].

Consistent with our previous analyses of PKA-C[L205R], the allosteric network of PKA-C[DNAJB1] exhibits an overall attenuation in the density of correlations across the kinase with a marked reduction near the nucleotide-binding site and substrate-binding groove. The introduction of community-specific analyses assists in deciphering how the fusion of the J-domain perturbs the allosteric network and further underscores the linkage between the reduced binding cooperativity and the decrease in cross-talk among the structural communities[32]. The coupling between ComH, responsible for the docking of proteins such as PKI or R-subunits, to other communities is completely ablated, explaining the lower degree of binding cooperativity observed for PKA-C[DNAJB1], and further suggesting that the R/C-interactions may be altered. Moreover, the dramatic decrease in the inter-community coupling of ComC, adjacent to the αA-helix, reflects the structural and/or dynamic changes caused by the J-domain fusion to the αA-helix. Overall, the similarities observed between PKA-C[L205R] and PKA-C[DNAJB1] suggest a common mechanism, involving concomitant reductions in binding cooperativity and intramolecular allostery that may elicit their aberrant function.

Indeed, both computational and experimental methods have successfully elucidated allosteric pathways in a variety of different systems outside of protein kinase A[45–52]. Perhaps more importantly, many of these studies have shown how disruptions in these pathways act to change function. Taken with our previous studies, nucleotide/substrate-binding cooperativity emerges as a critical function in protein kinase A that arises from the structural changes of multiple domains (i.e., communities) that must be coordinated and synchronized to ensure a cooperative binding response[18,20]. Like other tumorigenic transcriptomes that result in a fully active PKA-C, it is likely that alterations in the allosteric network impart defective binding cooperativity and may play a role in their aberrant functions[35,36,53]. As for the chimeric PKA-C[DNAJB1], attenuated or dysfunctional responses to allosteric effectors (such as ATP) may result in defective spatial and temporal control of phosphorylation-mediated signaling in other protein kinases, leading to disease.

## Methods
### Sample preparation of PKA-C[WT] and PKA-C[DNAJB1].
Recombinant PKA-C[WT] was expressed and purified as described in Walker et al.[20]. PKA-C[DNAJB1] was cloned into a pET-28a (+) vector. A tobacco etch virus cleavage site was incorporated via mutagenesis into the vector between the cDNA coding for the kinase and a thrombin cleavage site. Transformed *E. coli BL21 (DE3) pLyss* (Agilent) cells were cultured in M9 minimal media supplemented with $^{15}NH_4Cl$. Protein overexpression was induced with 0.4 mM isopropyl β-D-thiogalactopyranoside (IPTG) and carried out overnight at 20 °C. Following harvest, the cell pellet was resuspended in 50 mM Tris-HCl, 30 mM $KH_2PO_4$, 300 mM NaCl, 5 mM 2-mercaptoethanol, 0.15 mg/mL Lysozyme, 200 μM ATP, DNaseI, Roche ethylenediaminetetraacetic acid (EDTA)-free protease inhibitor tablet (pH 8.0) and lysed using French press. The solution was cleared by centrifugation (16,000 rpm, 4 °C, 45 min) and the supernatant incubated with $Ni^{2+}$ nitrilotriacetic acid resin (Thermo Scientific, 1 ml of resin per liter of culture) at 4 °C overnight. The resin was then washed with 50 mM Tris-HCl, 30 mM $KH_2PO_4$, 300 mM NaCl, 5 mM 2-mercaptoethanol, 0.5 mM phenylmethylsulfonyl fluoride (PMSF) (pH 8.0). The resin was further washed with the same buffer supplemented with 10 mM imidazole. PKA-C[DNAJB1] was eluted using the same buffer with 250 mM imidazole. Sodium dodecyl sulphate polyacrylamide gel electrophoresis (SDS–PAGE) electrophoresis samples were taken at each purification step listed and shown in Supplementary Fig. 8a. Fractions containing PKA-C[DNAJB1] were cleaved for 18 h at 4 °C in a dialysis buffer composed of 20 mM $KH_2PO_4$, 25 mM KCl, 5 mM 2-mercaptoethanol, 0.1 mM PMSF (pH 7.0). The phosphorylation states of PKA-C[DNAJB1] were separated using a cation exchange column (HiTrap Q-SP, GE Healthcare Life Sciences using a linear gradient of KCl in 20 mM $KH_2PO_4$.

**Sample preparation of DNAJB1.** The 69 amino-acid sequences corresponding to the DNAJB1 heat-shock protein fragment, DNAJB1[1–69], was cloned into the pET-28a (+) vector. DNAJB1[1–69] was expressed in *E. coli BL21 (DE3)* (Agilent) in M9 minimal media containing $^{15}NH_4Cl$ and $^{13}C$-glucose as the sole source of nitrogen and carbon, respectively. Overexpression was induced with 0.4 mM IPTG and carried out for 5 h at 30 °C. The cell pellet was resuspended in 20 mM Tris-HCl, 300 mM NaCl, 5 mM 2-mercaptoethanol, 15 mg/mL Lysozyme, DNaseI, Roche EDTA-free protease inhibitor tablet (pH 8.0) and lysed using sonication. The lysate was cleared by centrifugation (18,000 rpm, 4 °C, 30 min,) and the supernatant was incubated with $Ni^{2+}$ nitrilotriacetic acid resin at 4 °C overnight. Resin was washed with 20 mM Tris-HCl, 300 mM NaCl, 5 mM 2-mercaptoethanol, 0.5 mM PMSF (pH 8.0). DNAJB1[1–69] was eluted using the same buffer supplemented with 250 mM imidazole. SDS–PAGE electrophoresis samples were taken at each purification step listed as shown in Supplementary Fig. 8b. The elution was dialyzed for 3 h at 4 °C into 20 mM Tris-HCl, 150 mM NaCl and 5 mM 2-mercaptoethanol (pH 8.0). To cleave the His-tag, an appropriate amount of thrombin was added to the solution and allowed to occur at RT for 4 h. The cleavage reaction was monitored by SDS–PAGE, and upon completion, thrombin was inactivated by adding 1 mM PMSF. To purify the protein, the cleavage solution was passed through a $Ni^{2+}$ nitrilotriacetic acid resin. The flowthrough was collected, concentrated using a 10 kDa, and subsequently a 3 kDa spin concentrator and stored at 4 °C in 20 mM Tris-HCl, 150 mM NaCl and 2 mM DTT supplemented with 0.5 mM of PMSF (pH 8.0).

**Sample preparation of peptides.** Kemptide (LRRASLG), CREB ([123]KRREILSRRPSYR[135]), and KSR1 ([880]LPKLNRRLSHPGHFWKS[896]), and PKI[5–24] were synthesized on a CEM Liberty Blue microwave synthesizer. All peptides were purified using reverse-phase high-pressure liquid chromatography. The purified peptide was concentrated, lyophilized, and stored at −20 °C. All peptides molecular weight was verified by MALDI-TOF. Kemptide's molecular weight quantity was verified by amino-acid analysis (Texas A&M University).

**ITC measurements.** ITC measurements were performed with a low-volume NanoITC (TA Instruments). PKA-C[WT] and PKA-C[DNAJB1] were dialyzed into 20 mM MOPS, 90 mM KCl, 10 mM DTT, 10 mM $MgCl_2$, and 1 mM $NaN_3$ (pH 6.5). PKA-C[WT] and PKA-C[DNAJB1] concentrations for ITC measurements were between 80 and 130 μM as confirmed by $A_{280} = 53,860$ $M^{-1}$ $cm^{-1}$ and $A_{280} = 62,800$ $M^{-1}$ $cm^{-1}$, respectively. Approximately 300 μL of the kinase was used for each experiment, with 50 μL of 2 mM ATPγN and/or 0.6–1 mM PKI[5–24] in the titrant syringe. The heat of dilution of the ligand into the buffer was taken into account for all experiments and subtracted. For experiments with saturated nucleotide, 2 mM ATPγN was added. All measurements were performed at 300 K in triplicates. The binding was assumed to be 1:1, and curves were analyzed with the NanoAnalyze software (TA Instruments) using the Wiseman Isotherm[43]:

$$\frac{d[MX]}{d[X_{tot}]} = \Delta H° V_0 \left[ \frac{1}{2} + \frac{1 - \frac{1-r}{2} - R_m/2}{(R_m^2 - 2R_m(1-r) + (1+r)^2)^{1/2}} \right] \quad (1)$$

where d[MX] is the change in total complex relative to the change in total protein concentration, $d[X_{tot}]$ is dependent on $r$, the ratio of $K_d$ relative to the total protein concentration, and $R_m$, the ratio between total ligand and total protein concentration. The free energy of binding was determined using the following:

$$\Delta G = RT \ln K_d$$

where $R$ is the universal gas constant, and $T$ is the temperature at measurement

(300 K). The entropic contribution to binding was calculated using the following:

$$T\Delta S = \Delta H - \Delta G.$$

The degree of cooperativity ($\sigma$) was calculated as follows:

$$= \frac{K_{d\ Apo}}{K_{d\ Nucleotide}}$$

where $K_{d\ Apo}$ is the dissociation constant of $PKI_{5-24}$ binding to the apoenzyme, and $K_{d\ Nucleotide}$ is the corresponding dissociation constant for $PKI_{5-24}$ binding to the nucleotide-bound PKA-C.

**Enzyme assays.** Steady-state activity assays with Kemptide, CREB, and KSR1 were performed under saturating ATP concentrations and spectrophotometrically at 298 K as described by Cook et al.[44]. The values of $V_{max}$ and $K_M$ were obtained from a non-linear fit of the initial velocities to the Michaelis–Menten equation.

**Small-angle X-ray scattering.** Following the purification method outlined above, PKA-C$^{DNAJB1}$ was buffer exchanged into 20 mM MOPS, 90 mM KCl, 60 mM MgCl$_2$, 10 mM DTT, and 1 mM NaN$_3$ (pH 6.5) and concentrated to 70 μM. For binary and ternary samples, 12 mM ATPγN and/or 70 μM PKI (1:1 ratio) was added to PKA-C$^{DNAJB1}$. The fitting of structures from MD simulations against the experimental SAXS curves was performed using the FoXS web server[54]. One hundred random snapshots from MD simulations for PKA-C$^{WT}$ and PKA-C$^{DNAJB1}$ were chosen as representative structures for the conformational ensembles, and the fitting of the SAXS data was carried out for a range of $q$ values spanning from 0.01 to 0.2.

**NMR spectroscopy.** Uniformly $^{15}$N-labeled PKA-C$^{WT}$ and PKA-C$^{DNAJB1}$ were overexpressed and purified as described above. NMR experiments were performed in 90 mM KCl, 20 mM KH$_2$PO$_4$, 10 mM dithiothreitol (DTT), 60 mM MgCl$_2$, and 1 mM NaN$_3$ at pH 6.5. Standard [$^1$H-$^{15}$N]-TROSY-HSQC experiments were carried out for PKA-C$^{WT}$ and PKA-C$^{DNAJB1}$ on an 850-MHz Bruker Advance III spectrometer equipped with a TCI cryoprobe, respectively. Concentrations for samples were 0.1–0.3 mM as determined by A$_{280}$ measurements, 12 mM ATPγN or ADP was added for the nucleotide-bound form, and 0.2–0.4 mM PKI$_{5-24}$ for the ternary complex. Spectra were collected at 300 K, processed using NMRPipe[55], and visualized using Sparky[56].

All [$^1$H-$^{15}$N]-TROSY-HSQC experiments were acquired with 2048 (proton) and 256 (nitrogen) complex points. Combined CSPs were calculated using $^1$H and $^{15}$N chemical shifts according to the following:

$$\Delta\delta = \sqrt{(\Delta\delta H)^2 + (0.154 \times \Delta\delta N)^2} \tag{2}$$

**Backbone resonance assignment of DNAJB1$_{1-69}$.** Uniformly $^{13}$C,$^{15}$N-labeled DNAJB1$_{1-69}$ was overexpressed and purified as described above. NMR experiments were performed in 90 mM KCl, 20 mM KH$_2$PO$_4$, 10 mM DTT, 60 mM MgCl$_2$, and 1 mM NaN$_3$ (pH 6.5). Standard 3D triple resonance experiments were carried out for DNAJB1$_{1-69}$ on a 600 MHz Bruker Avance NEO spectrometer equipped with a triple resonance cryogenic probe at 300 K. Concentrations for samples were 0.3 mM as determined by A$_{280}$ measurements. All NMR data were processed using NMRPipe[55] and visualized using Sparky[56].

**Chemical shift analyses**

*COordiNated ChemIcal Shift bEhavior (CONCISE).* CONCISE was used to measure the change in equilibrium position using the following PKA-C constructs: apo, ATPγN, ADP, and ATPγN/PKI$_{5-24}$. This method identifies sets of residues whose chemical shifts respond linearly to the conformational transition using principal component analysis. Community-based and domain-based CONCISE analyses use similar methods using only select residues that comprise either an individual community or domain as defined by McClendon et al.[32,57]. ΔΔG values from NMR derived from CONCISE analysis and based on the extent of closure (% closed). Values used for the calculations can be found in Supplementary Table 4. PKA-C$^{WT}$ ITC values were used as a reference with apo PKA-C$^{WT}$: ΔΔG = 0 and 0% closed; and +ATPγN/PKI-bound PKA-C$^{WT}$: ΔΔG = −15 and 100% closed. The error was calculated by using the standard deviation of the gaussian fits from CONCISE analysis.

*CHEmical Shift Covariance Analysis (CHESCA).* CHESCA was used to identify and functionally characterize allosteric networks of residues eliciting concerted responses to, in this case, nucleotide and pseudo-substrate. A total of four states were used to identify inter-residue correlations: apo, ATPγN-bound, ADP-bound, and ATPγN/PKI$_{5-24}$. The identification of inter-residue correlations by CHESCA relies on agglomerative clustering and singular value decomposition. Pairwise correlations between chemical shift variations experienced by different residues were calculated to identify networks. When plotted on a correlation matrix, this allows for the identification of regions that are correlated to one another. For each

residue, the max change in chemical shift was calculated in both the $^1$H ($x$) and $^{15}$N ($y$) dimensions ($\Delta\delta_{x,y}$). Residues were included in CHESCA analysis only if they satisfied the following: $\Delta\delta_{x,y} > \frac{1}{2}\Delta\nu_{xA,yA} + \frac{1}{2}\Delta\nu_{xB,yB}$, where $A$ and $B$ correspond to two different forms analyzed. (Note that there is no dependence on which two forms satisfied this statement) For every residue $x$, correlation scores were calculated by finding the total no. of residues correlated (with $R_{ij} > 0.98$) to residue $x$ and dividing by the total no. of residues in the kinase (350). Community CHESCA analysis is a chemical shift-based correlation map between various functional communities within the kinase. Each community is a group of residues (McClendon et al.[35]) associated with a function or regulatory mechanism. Mathematically, this community-based CHESCA analysis is a selective interpretation of CHESCA, where we evaluate a correlation score between residues in various communities, as shown below. To represent community-based CHESCA analysis, we lowered the correlation cutoff such that $R_{cutoff} > 0.8$. Suppose community X and community Y has $n_X$ and $n_Y$ number of assigned residues respectively, the correlation score between X and Y is defined as,

$$R_{X,Y} = \text{Number of } \left(R_{ij} > R_{cutoff}\right) / (n_X * n_Y).$$

Where $R_{ij}$ is the CHESCA correlation coefficient between residue $i$ (belongs to community A) and residue $j$ (belongs to community B). $R_{cutoff}$ is the correlation value cutoff. $R_{X,Y}$ can assume values ranging from 0 (no correlation between residues in X and Y) to 1 (all residues in X are correlated with all residues in Y).

**MD simulations.** Parallel MD simulations were set up to compare PKA-C$^{WT}$ and PKA$^{DNAJB1}$. These simulations were repeated for the apo, the binary (ATP-bound), and the ternary complexes (ATP/PKI$_{5-24}$). The systems with PKA-C$^{WT}$ and PKA-C$^{DNAJB1}$ were built directly from the X-ray structures starting from the closed configurations (PDB ID: 1ATP and 4WB7, respectively)[12,58]. All simulations were performed using GROMACS 4.6[59] in the CHARMM36 force field[60]. The all-atom structures were solvated in a rhombic dodecahedron solvent box with a TIP3P[61] water molecule layer extending ~10 Å away from the surface of the proteins. Counter ions (K$^+$ and Cl$^-$) were added to ensure electrostatic neutrality corresponding to an ionic concentration of ~150 mM. The LINCS[62] algorithm was applied to constrain all covalent H-bonds to the equilibrium length, and particle-mesh Ewald[63] was used to treat long-range electrostatic interactions with a real-space cutoff of 10 Å. All systems were minimized using steepest descent algorithm, and then were gradually heated to 300 K at a constant volume over 1 ns, using harmonic restraints with a force constant of 1000 kJ/(mol*Å$^2$) on heavy atoms of both proteins and nucleotides. Over the following 12 ns of simulations at constant pressure (1 atm) and temperature (300 K), the restraints were gradually released. The systems were equilibrated for an additional 20 ns without positional constraints. A Parrinello–Rahman barostat[64] was used to keep the pressure constant, while a V-rescale thermostat with a time step of 2 fs was used to maintain a constant temperature. Each system was simulated for 1.05 μs, with snapshots recorded every 20 ps. A total of 6.3 μs and 315,000 conformations were utilized for the analyses.

**Statistics and reproducibility.** Numerical ΔG, ΔH, −TΔS, and $K_d$ values shown in Supplementary Tables 1 and 2 are averages of three independent ITC experiments with errors calculated as the standard deviation between these three measurements. Numerical values of $V_{max}$ and $K_M$ were determined with a non-linear regression function assuming Michaelis-Menten kinetics for substrate concentration versus velocity using GraphPad Prism 8.0.1. Error in $k_{cat}/K_M$ values was propagated from error in $k_{cat}$ and $K_M$.

**Reporting summary.** Further information on research design is available in the Nature Research Reporting Summary linked to this article.

## Data availability

Raw data from steady-state activity assays and corresponding analysis is deposited on the Data Repository for the University of Minnesota (DRUM) and can be found at the following link: http://hdl.handle.net/11299/217205. Proton and amide chemical shift list files of wild-type PKA-C and the chimeric mutant (PKA$^{DNAJB1}$) are deposited on DRUM and can be found at the following link: http://hdl.handle.net/11299/217206. The source data for the graphs in the main figures are provided in Supplementary Data 1. All other data sets generated during and/or analyzed during the current study are available from the corresponding author on reasonable request.

## Code availability

MATLAB scripts used for the chemical shift analyses presented are deposited on DRUM and can be found at the following link: http://hdl.handle.net/11299/217206.

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

## Acknowledgements

This work was supported by the National Institutes of Health, GM100310 (G.V.), S10 OD021536 (G.V.), HL139065 (D.D.T.), and AG26160 (D.D.T); and the American Heart Association, 20PRE35120253 (C.W.). NMR experiments were carried out at the Minnesota NMR Center and MD simulations using the Minnesota Supercomputing Institute. The SAXS instrument at the University of Utah was funded by the Department of Energy (Dr. J. Trewhella).

## Author contributions

C.O. and C.W. collected and analyzed activity assays, NMR, and ITC data and contributed to the writing of the manuscript. A.K carried out preliminary experiments. Y.W. analyzed SAXS data and completed MD simulations. D.K.B collected the SAXS data. M.V.S. wrote all scripts for CHESCA and CONCISE analyses. F.P., D.D.T., D.A.B., S.M.S., and S.S.T. contributed to the critical analysis of the data and writing of the manuscript. G.V. conceived and directed the project, along with helping with data analysis and writing the manuscript.

## Competing interests

The authors declare no competing interests.
