## [Peer Review File · Communications Biology]

Reviewers' comments:

Reviewer #1 (Remarks to the Author):

This structural study focuses on the chimeric protein kinase PKA-CDNAJB1 that has been identified as being fully functional in the liver cancer fibrolamellar hepatocellular carcinoma (FL-HCC). The crystal structure of PKA-CDNAJB1 has been known for some time and has been the topic of other papers. The catalytic core of the enzyme appears to be indistinguishable from the native cAMP-dependent protein kinase A (PKA-C). The authors suggest that the chimera's allosteric response to nucleotide and pseudo-substrate binding is subtly altered from PKA-C.

These are highly specialized findings that represent, at best, an incremental advance in our appreciation of PKA-CDNAJB1. The work is highly technical and offers very few insights into how the chimera may serve as a primary driver of FL-HCC. This article does not offer the degree of novelty or scientific advance that is necessary to publish in Nature Communications.

Other limitations to this work include:

1) Why is the SAXS data presented in figure 2A-C which is relevant to the interaction with PKI 5-24 not included with Figure 1. The data in figure 1C showing dissociation constants for PKI is potentially more interesting and easier to understand if the data from Figure 2A-C is included.

2) Experiments in figure 1B using the Kemptide (a 7 amino acid peptide) are interesting but show a very small effect. It would be more meaningful if these studies were expanded to survey a range of substrates. This is important because of the kinetics of cAMP-dependent protein kinase phosphorylation varies depending upon the substrate. As does PKI inhibition.

3) A rather complicated argument on page 6 is laid out to suggest that PKI5-24 binds the PKA-CDNAJB1/ATP γ N binary complex with a 7-fold decrease in binding affinity. The authors state that they are showing favorable enthalpic and unfavorable entropic contributions. The authors conclude that a 10-fold reduction in σ (Table S3), suggesting the J-domain marginally influences PKI5-24 binding affinity; but it causes a sizable decrease in nucleotide/pseudo-substrate binding cooperativity. This data is hard to follow, particularly for a general audience that is the target for Nature Communications. This section needs to be simplified and clarified.

4) On page 8 the authors derive a CHESCA matrix for PKA- CDNAJB1. They conclude that the extensive rewiring of the intramolecular communication is even more apparent when they combine the CHESCA with the definition of the structural and functional communities. This is a very confusing and unclear statement that seems to say nothing. More clarification and additional studies are necessary. For example are there differences between PKA- CDNAJB1 and any of the Cushing's syndrome mutants (PKA-C L206R) mentioned in the text.

4) This is a purely biochemical study, yet the authors conclude that the subtle differences that they observe in kinetics translate into reduced allosteric cooperativity may alter kinase's regulation, substrate recognition, and nuclear export, leading to tumorigenesis. There is no data in this article that points to changes in nuclear export of PKA-CDNAJB. The authors must provide some cellular data that supports this conclusion.

Reviewer #2 (Remarks to the Author):

Overall, Olivieri et al have submitted a well written manuscript describing how the fusion of the J-domain of DNAJB1 to the N-terminus of PKA-C, in place of exon 1 derived residues, has altered the allosteric response of the mutant PKA-CDNAJB1 enzyme to nucleotide/PKI binding compared to the

wild type enzyme. The mutant PKA-CDNAJB1 is seen as the primary driver of FL-HCC and more recently has been implicated in other neoplasms. Supported by complementary approaches of ITC, SAXS, NMR spectroscopy, and MD, the authors have convincingly determined attenuated positive binding cooperativity between nucleotide and the PKI peptide. In further using computational methods to analyze the NMR data, the authors have identified changes in the intramolecular coupling of " structural communities" as the mechanism underlying these changes in the protein. These results are presented in a visually clear manner and these insights contribute towards our understanding of this elusive disease.

In the greater context of disease, the pseudo-substrate PKI used in the described experiments is interpreted as a proxy for endogenous regulatory subunits and other substrates. The authors believe the described dysfunctional binding cooperativity of PKA- CDNAJB1 seen with PKI may lead to an altered interactome and phosphoproteome, and thus contribute to disease pathogenesis in concert with other documented effects of the mutation. Although this is a defensible assertion, this paper could be stronger if some binding studies were performed with regulatory subunits or other known protein substrates; however these experiments are not critical to understanding changes at the protein level but would provide more insight into the disease.

I only have minor points to address/discuss:

1. The authors have previously published a similar study involving the L205R mutant of PKA- C linked to Cushing's syndrome. But outside of PKA-C, are there other known examples of disease states specifically caused by a defective allosteric network in a protein kinase? If so, it would add strength to mention other examples in the discussion.
2. It is very interesting how the presence of the J-domain can significantly alter binding cooperativity at the nucleotide binding site located toward the other end of the molecule. As the J-domain has been shown to be fairly flexible, it would make sense that the intramolecular communication between the various structural communities within the protein may be very sensitive to external forces. Would we expect that protein binding partners interacting with PKA-C with a significant surface footprint would also alter the allosteric properties of the enzyme?
3. In the disease state it is known that PKA-CDNAJB1 is overexpressed and may be mislocalized. In general, are there any indications with this kinase system, or perhaps another kinase system, how allosteric disruptions of this magnitude and nature compare to significant overexpression of a kinase in relation to changing its phosphoproteome?
4. In Figure 2, it is difficult to distinguish the fitted curves in panels A, B, and C from each other. Perhaps the curves can be drawn with thicker lines or different color combinations can be used. At the very least more contrasting colors should be chosen for the different J- domain conformations as they all appear to be a similar shade.

RESPONSE TO REVIEWS

Reviewer #1

“This structural study focuses on the chimeric protein kinase PKA-CDNAJB1 that has been identified as being fully functional in the liver cancer fibrolamellar hepatocellular carcinoma (FL-HCC). The crystal structure of PKA-CDNAJB1 has been known for some time and has been the topic of other papers. The catalytic core of the enzyme appears to be indistinguishable from the native cAMP-dependent protein kinase A (PKA-C). The authors suggest that the chimera's allosteric response to nucleotide and pseudo-substrate binding is subtly altered from PKA-C.

These are highly specialized findings that represent, at best, an incremental advance in our appreciation of PKA-CDNAJB1. The work is highly technical and offers very few insights into how the chimera may serve as a primary driver of FL-HCC. This article does not offer the degree of novelty or scientific advance that is necessary to publish in Nature Communications.”

AU: We thank this reviewer for the careful reading of our manuscript. We respectfully disagree with her/his overall assessment of the paper. We would also like to point out that the difference in allosteric cooperativity between the wild-type and chimeric kinase is not subtle (as implied in this review), but 13-fold (see Table S2). Note that in the previous version of the main text, the value was mistyped (10-fold) but correct in Table S2. These changes in cooperativity coefficients are indeed substantial and modulate protein/ligand and macromolecular assembly and recognition (see Williamson doi:10.1038/nchembio.102 and Bonin et al. doi:10.1016/j.bpj.2019.08.015).

As stated by this reviewer, the catalytic core of the wild-type enzyme and the chimera are virtually indistinguishable, yet their biological roles are different. Therefore, the aberrant function is the product of small, but significant changes in the chimera's structural dynamics that are not visible by X-ray. This phenomenon is not limited to the present study. Other systems (now cited in the revised manuscript) show that marginal structural/dynamic changes are accountable for the enzymes' aberrant function.

In the case of PKA-C^{DNAJB1}, these changes have been hypothesized by Sandford and coworkers in a recent molecular dynamics study (Ref. 22). More importantly, disruption of allosteric cooperativity has been one of the main causes of dysfunctional interactome or phosphoproteome (see Cushing's mutations, Ref. 20), and it is likely correlated to the variations of the phosphoproteome identified by Scott and coworkers (1)

In this revised version, we addressed the limitation suggested by this reviewer and improved our manuscript based on her/his critique. Below are the point-by-point responses to her/his concerns.

REV#1: 1) Why is the SAXS data presented in figure 2A-C which is relevant to the interaction with PKI 5-24 not included with Figure 1. The data in figure 1C showing dissociation constants for PKI is potentially more interesting and easier to understand if the data from Figure 2A-C is included.

AU: We agree with this comment and have now combined Figures 1 and 2 into a single figure (Figure 1) to emphasize the connection between biochemical results and SAXS analyses. We have reorganized the text as well to follow this new layout.

REV#1: 2) Experiments in figure 1B using the Kemptide (a 7 amino acid peptide) are interesting but show a very small effect. It would be more meaningful if these studies were expanded to survey a range of substrates. This is important because of the kinetics of cAMP-dependent protein kinase phosphorylation varies depending upon the substrate. As does PKI inhibition.

AU: Following this reviewer's suggestion, we now survey a range of substrates, including a common substrate of PKA, CREB (cAMP response element-binding protein), and a substrate that has been found to be hyper-phosphorylated in FL-HCC, KSR1 (kinase suppressor of Ras 1) (1). These experiments are now included in Figure 2, Figure S3, and Table S3, and discussed in lines 126-134. These new data corroborate our original conclusions.

REV#1: 3) A rather complicated argument on page 6 is laid out to suggest that PKI5-24 binds the PKA- CDNAJB1/ATP γ N binary complex with a 7-fold decrease in binding affinity. The authors state that they are showing favorable enthalpic and unfavorable entropic contributions. The authors conclude that a 10-fold reduction in σ (Table S3), suggesting the J-domain marginally influences PKI5-24 binding affinity; but it causes a sizable decrease in nucleotide/pseudo-substrate binding cooperativity. This data is hard to follow, particularly for a general audience that is the target for Nature Communications. This section needs to be simplified and clarified.

AU: We agree. The paragraph describing the changes in binding cooperativity elicited by PKA-C^{DNAJB1} has been simplified and shortened for the general audience of Communications Biology.

REV#1: 4) On page 8 the authors derive a CHESCA matrix for PKA- CDNAJB1. They conclude that the extensive rewiring of the intramolecular communication is even more apparent when they combine the CHESCA with the definition of the structural and functional communities. This is a very confusing and unclear statement that seems to say nothing. More clarification and additional studies are necessary. For example are there differences between PKA- CDNAJB1 and any of the Cushing's syndrome mutants (PKA-C L206R) mentioned in the text.

AU: We agree. We have made some changes in this section's wording, clarifying the CHESCA analyses' results. In the discussion section, we now emphasize the connection between our new findings and our previous study of L205R.

REV#1: 5) This is a purely biochemical study, yet the authors conclude that the subtle differences that they observe in kinetics translate to reduced allosteric cooperativity may alter kinase's regulation, substrate recognition, and nuclear export, leading to tumorigenesis. There is no data in this article that points to changes in nuclear export of PKA-CDNAJB. The authors must provide some cellular data that supports this conclusion.

AU: We agree. Accordingly, we have toned down our claims, and we have removed mention of specific cellular processes that could be altered from our studies. Indeed, we speculate that these observed changes contribute to dysregulation of PKA-C^{DNAJB1} in FL-HCC.

Reviewer #2:

Overall, Olivieri et al have submitted a well written manuscript describing how the fusion of the J-domain of DNAJB1 to the N-terminus of PKA-C, in place of exon 1 derived residues, has altered the allosteric response of the mutant PKA-CDNAJB1 enzyme to nucleotide/PKI binding compared to the wild type enzyme. The mutant PKA-CDNAJB1 is seen as the primary driver of FL-HCC and more recently has been implicated in other neoplasms. Supported by complementary approaches of ITC, SAXS, NMR spectroscopy, and MD, the authors have convincingly determined attenuated positive binding cooperativity between nucleotide and the PKI peptide. In further using computational methods to analyze the NMR data, the authors have identified changes in the intramolecular coupling of “ structural communities” as the mechanism underlying these changes in the protein. These results are presented in a visually clear manner and these insights contribute towards our understanding of this elusive disease.

In the greater context of disease, the pseudo-substrate PKI used in the described experiments is interpreted as a proxy for endogenous regulatory subunits and other substrates. The authors believe the described dysfunctional binding cooperativity of PKA- CDNAJB1 seen with PKI may lead to an altered interactome and phosphoproteome, and thus contribute to disease pathogenesis in concert with other documented effects of the mutation. Although this is a defensible assertion, this paper could be stronger if some binding studies were performed with regulatory subunits or other known protein substrates; however these experiments are not critical to understanding changes at the protein level but would provide more insight into the disease.

I only have minor points to address/discuss:

AU: We thank this reviewer for his/her positive and constructive comments. Below are the point-by-point responses to his/her concerns.

REV#2: 1. The authors have previously published a similar study involving the L205R mutant of PKA- C linked to Cushing’s syndrome. But outside of PKA-C, are there other known examples of disease states specifically caused by a defective allosteric network in a protein kinase? If so, it would add strength to mention other examples in the discussion.

AU: We augmented the Discussion section (lines 291-294) emphasizing the successful elucidation of allosteric pathways and how disease mutations often affect these pathways citing appropriate literature.

REV#2: 2. It is very interesting how the presence of the J-domain can significantly alter binding cooperativity at the nucleotide binding site located toward the other end of the molecule. As the J-domain has been shown to be fairly flexible, it would make sense that the intramolecular communication between the various structural communities within the protein may be very sensitive to external forces. Would we expect that protein binding partners interacting with PKA-C with a significant surface footprint would also alter the allosteric properties of the enzyme?

AU: Yes. Giving the flexibility of the J-domain, this was surprising to us as well. At the moment, we can speculate that the dynamics (motions) of the enzyme can be more affected by the J-domain. Currently, we are performing the initial experiments to test this hypothesis. However, it has been difficult (as for everybody) to carry out effective

research in the laboratory. Regarding larger interacting binding partners, we have not tested this. As inferred from the reviewer's question, we too anticipate that there are stronger effects based on the footprint of the interacting partner.

REV#2: 3. In the disease state it is known that PKA-CDNAJB1 is overexpressed and may be mislocalized. In general, are there any indications with this kinase system, or perhaps another kinase system, how allosteric disruptions of this magnitude and nature compare to significant overexpression of a kinase in relation to changing its phosphoproteome?

AU: To date, there are no reports on mislocalization of the liver chimera. Regarding the effects of overexpression vs. changing phosphoproteome, enzyme overexpression does not necessarily coincide with the loss in substrate fidelity. Usually, overexpression amplifies the existing signaling pathways, whereas disruption of the recognition site often leads to impaired phosphorylation or change in fidelity.

REV#2: 4. In Figure 2, it is difficult to distinguish the fitted curves in panels A, B, and C from each other. Perhaps the curves can be drawn with thicker lines or different color combinations can be used. At the very least more contrasting colors should be chosen for the different J-domain conformations as they all appear to be a similar shade.

AU: We agree and have updated the colors used in this figure (now Figure 1 and S1) to emphasize the fitting of the SAXS data.

REFERENCES

- (1) Turnham, R.E. *et al.* An acquired scaffolding function of the DNAJ-PKAc fusion contributes to oncogenic signaling in fibrolamellar carcinoma. *eLife* **8**, e44187, doi:10.7554/eLife.44187 (2019).

REVIEWERS' COMMENTS:

Reviewer #1 (Remarks to the Author):

It have gone over the revised manuscript and remain skeptical about the value of this work. The authors conclude that the DNAJ-PKA fusion has a 13 fold difference in its allosteric binding cooperativity for nucleotide and PKI. Yet this term is not adequately defined.

In addition since PKI is a nucleotide dependent competitive inhibitor of PKAc one would think that the fusion kinase would have different kinetics for substrates. Yet the new data in figure 2 suggest that native PKA and DNAJ-PKAc have the same substrates selectivity and kinetics for three substrates. This result seems to underscore the rather esoteric finding of this work.

I cannot recommend publication.

Reviewer #2 (Remarks to the Author):

The changes made in the revised copy of the manuscript are appropriate. The figures are more clear and the text is now easier to follow for a general audience. The additional experimental results add strength and are reassuring. I have no additional constructive criticism to offer and I believe this manuscript is of excellent caliber and should be published.

RESPONSES TO REVIEWERS

Reviewer #1:

“It have gone over the revised manuscript and remain skeptical about the value of this work. The authors conclude that the DNAJ-PKA fusion has a 13 fold difference in its allosteric binding cooperativity for nucleotide and PKI. Yet this term is not adequately defined.

In addition since PKI is a nucleotide dependent competitive inhibitor of PKAc one would think that the fusion kinase would have different kinetics for substrates. Yet the new data in figure 2 suggest that native PKA and DNAJ-PKAc have the same substrates selectivity and kinetics for three substrates. This result seems to underscore the rather esoteric finding of this work. I cannot recommend publication.”

AU: We thank this reviewer for his/her time to review our work. We further clarified the definition of allosteric binding cooperativity between nucleotide and substrates, specifying that it is a **K-type cooperativity**. We showed a 13-fold difference in binding cooperativity between nucleotide and PKI that is likely to affect the holoenzyme's assembly as PKI contains a homologous recognition sequence of the R subunits.

In this revised version on page 5, we added the following sentence:

“Previous studies from our group have shown that PKA-C^{WT} behaves like a K-type enzyme, exhibiting positive binding cooperativity between nucleotide (ATP γ N) and pseudo-substrate (PKI₅₋₂₄). The binding cooperativity can be reduced by mutations, while maintaining a virtually constant maximal rate^{20,23,24}”

As suggested by this reviewer in the first round of revision, we repeated the kinetic experiments with different substrates available for PKA-C^{DNAJB1}, and, as expected, we did not observe significant changes in Vmax for the substrates analyzed. Indeed, this is typical of allosteric enzymes with **K-type** cooperativity in which the binding constants may vary, but the Vmax of the enzyme is virtually unaltered. In contrast, enzymes with a **V-type** cooperativity are expected to change binding affinity and Vmax (see *Allosteric Regulatory Enzymes* by Thomas Traut, Springer 2008).

Reviewer #2:

“The changes made in the revised copy of the manuscript are appropriate. The figures are more clear and the text is now easier to follow for a general audience. The additional experimental results add strength and are reassuring. I have no additional constructive criticism to offer and I believe this manuscript is of excellent caliber and should be published.”

AU: We would like to thank this reviewer for the time spent on reviewing our manuscript during these challenging times. His/her comments and critiques have helped to improve our paper.